# Optimal Efficacy and Safety of Humanized Anti-Scg3 Antibody to Alleviate Oxygen-Induced Retinopathy

**DOI:** 10.3390/ijms23010350

**Published:** 2021-12-29

**Authors:** Ye He, Hong Tian, Chang Dai, Rong Wen, Xiaorong Li, Keith A. Webster, Wei Li

**Affiliations:** 1Department of Ophthalmology, Bascom Palmer Eye Institute, University of Miami School of Medicine, Miami, FL 33136, USA; yhe@doheny.org (Y.H.); daichang2018741032@outlook.com (C.D.); rwen@med.miami.edu (R.W.); kwebster@med.miami.edu (K.A.W.); 2Everglades Biopharma, LLC, Houston, TX 77054, USA; h.tian@evergladesbiopharma.com; 3Department of Ophthalmology, Cullen Eye Institute, Baylor College of Medicine, Houston, TX 77030, USA; 4Tianjin Key Laboratory of Retinal Functions and Diseases, Tianjin Branch of National Clinical Research Center for Ocular Disease, Eye Institute and School of Optometry, Tianjin Medical University Eye Hospital, Tianjin 300392, China; xiaorli@163.com

**Keywords:** retinopathy of prematurity, oxygen-induced retinopathy, secretogranin III, Scg3, anti-Scg3 therapy, anti-angiogenic therapy, humanized antibody, anti-VEGF, aflibercept, safety

## Abstract

The retinopathy of prematurity (ROP), a neovascular retinal disorder presenting in premature infants, is the leading causes of blindness in children. Currently, there is no approved drug therapy for ROP in the U.S., highlighting the urgent unmet clinical need for a novel therapeutic to treat the disease. Secretogranin III (Scg3) was recently identified as a disease-selective angiogenic factor, and Scg3-neutralizing monoclonal antibodies were reported to alleviate pathological retinal neovascularization in mouse models. In this study, we characterized the efficacy and safety of a full-length humanized anti-Scg3 antibody (hAb) to ameliorate retinal pathology in oxygen-induced retinopathy (OIR) mice, a surrogate model of ROP, by implementing histological and functional analyses. Our results demonstrate that the anti-Scg3 hAb outperforms the vascular endothelial growth factor inhibitor aflibercept in terms of efficacy and safety to treat OIR mice. Our findings support the development of anti-Scg3 hAb for clinical application.

## 1. Introduction

Retinopathy of prematurity (ROP) is a retinal vasoproliferative disorder primarily occurring in premature infants and is one of the leading causes of blindness in children [1]. The disease manifests with retinal vasculopathy, including regression of developing vessels or vaso-obliteration in Phase 1 with exposure to high oxygen and subsequent pathological retinal neovascularization (RNV) in Phase 2 due to relative hypoxia. In severe cases, the disease may progress to plus disease characterized by arterial tortuosity and venous dilation [1]. Current treatments for ROP include cryotherapy and laser photocoagulation for the ablation of avascular area. However, these treatments often damage the retina by destroying the peripheral vision in attempts to preserve the central vision and do not address the underlying cause of pathological RNV.

Vascular endothelial growth factor (VEGF) is a well-recognized key player in both physiological and pathological RNV in ROP infants [1]. Anti-VEGF drugs developed for other ocular neovascular diseases, such as wet age-related macular degeneration, diabetic retinopathy, and retinal vein occlusion, have been investigated for ROP therapy. Despite treatment efficacy, a potential concern is the safety of anti-VEGF drugs on the developing retina and other organs in premature infants. Various studies have reported that anti-VEGF drugs may cause significant vascular and macular abnormalities and serious adverse outcomes [2,3]. Intravitreal aflibercept (i.e., VEGF Trap) in neonatal mice and dogs suppressed retinal vascular development, disrupted retinal architecture, and reduced electroretinography (ERG) amplitudes [4,5,6]. Additionally, intravitreal anti-VEGF drugs can leak from the eye into the systemic circulation and reduce serum levels of VEGF, thereby affecting the development of other organs [7,8,9,10]. Indeed, clinical studies found that ROP infants treated with the anti-VEGF drug bevacizumab were associated with lower motor scores and higher rates of severe neurodevelopmental disability in comparison with laser treatment 18 months post treatment [11]. As a result, despite the recent approval of ranibizumab for ROP therapy in the European Union based on the RAINBOW study [12], there is currently no approved drug therapy for ROP in the United States. Thus, an urgent unmet clinical need is to develop an effective and safe anti-angiogenic therapy for the disease.

Using a unique technology of comparative ligandomics, we recently identified secretogranin III (Scg3) as a disease-selective angiogenic factor that preferentially induces angiogenesis of diseased but not healthy vessels in mice [13]. Our findings also demonstrated that Scg3-neutralizing monoclonal antibodies (mAbs) mitigated pathological RNV in oxygen-induced retinopathy (OIR) mice, a surrogate model of ROP, with minimal adverse effects on the developing retina and other organ systems [6,13]. These findings suggest that anti-Scg3 mAbs have the potential for optimal therapeutic efficacy and safety. In this study, we describe preclinical efficacy and safety of a full-length humanized anti-Scg3 antibody (hAb) in OIR mice.

## 2. Results

### 2.1. Alleviation of Pathological RNV in OIR Mice by Anti-Scg3 hAb

To determine the appropriate therapeutic dose, we generated a dose–response curve by intravitreally treating OIR mice with increased anti-Scg3 hAb at postnatal day 14 (P14) (Figure 1A–C). Analyses of retinal vessels stained with Alexa Fluor 488-conjugated isolectin B4 (AF488-IB4) at P17 revealed a decrease in pathological RNV in mice treated with anti-Scg3 hAb. The decrease correlated inversely with increased anti-Scg3 hAb in a dose-dependent manner, with maximal efficacy at 2 and 4 µg/eye. Interestingly, increasing doses of anti-Scg3 hAb resulted in a reduction in the central avascular area, rather than exacerbation of the vaso-obliteration, as would be predicted for anti-angiogenic drug therapies. The maximal decrease in central avascular area was observed at 2 and 4 µg/eye. We chose to further characterize anti-Scg3 hAb at the near maximal efficacy dose of 2 µg/eye.

We first compared the relative efficacies of anti-Scg3 hAb and aflibercept to ameliorate pathological RNV and central avascular area in OIR mice. Whereas doses of 2 µg/eye of either reagent conferred a significant reduction of pathological RNV and similar efficacy, only anti-Scg3 hAb treatment markedly reduced the central vaso-obliteration, suggesting that anti-Scg3 hAb not only inhibits pathological RNV but also promotes physiological angiogenesis of the developing retina (Figure 1D–F). As controls, phosphate-buffered saline (PBS) and control hIgG had minimal effects on OIR-induced RNV or the central avascular area.

To investigate the effects of anti-Scg3 hAb on severe OIR in vivo, we performed fluorescein angiography (FA) to quantify arterial tortuosity and venous dilation, which are two key manifestations of severe ROP. The results show that anti-Scg3 hAb and aflibercept alleviated both symptoms with similar efficacy (Figure 2).

### 2.2. Improved Visual Function of OIR Mice by Anti-Scg3 hAb Therapy

We evaluated the visual function of OIR retina by ERGs. Photopic flash electroretinography (pfERG) of OIR mice at P21 (7 days post intravitreal injection or DPI) demonstrated no changes of a-wave amplitude between groups (Figure 3A). However, intravitreal aflibercept reduced b-wave amplitude, corresponding to a decrease in visual function (Figure 3B). Anti-Scg3 hAb showed no statistical difference in b-wave amplitude relative to the PBS control group, indicating no adverse effects on visual function. pfERG analysis of OIR mice at P42 (28 DPI) also showed no change of a-wave amplitude between groups (Figure 3D). However, b-wave amplitude was again reduced by aflibercept but improved by anti-Scg3 hAb, corresponding to the deterioration and improvement of visual function, respectively (Figure 3E). Figure 3C,F shows representative ERG graphs from mice in each group at 7 and 28 DPI.

Scotopic flash electroretinography (sfERG) of OIR mice at P21 (7 DPI) demonstrated increased a-wave amplitude in anti-Scg3 hAb-treated mice at middle and high light intensities (Figure 3G), suggesting improvements in retinal function. In contrast, aflibercept treatment conferred no improvement over PBS. Whereas aflibercept and anti-Scg3 hAb bother conferred enhanced b-wave amplitude at the high light intensity, which is indicative of improved visual function, the effects of anti-Scg3 were more pronounced (Figure 3H). sfERG of OIR mice at P42 (28 DPI) detected a moderate increase in a-wave amplitude at the middle and high light intensities and a marked increase in b-wave amplitude at all light intensities in mice treated with anti-Scg3 hAb (Figure 3J,K). In contrast, while aflibercept treatment resulted in moderately increased b-wave amplitude at the high light intensity, we observed no effect at any other condition of sfERG. Figure 3I,L shows representative sfERG graphs for OIR mice from the same group with 1 cd·s/m^2^ light intensity at 7 and 28 DPI. Overall, anti-Scg3 conferred significant improvements in a-wave and b-wave amplitude at the middle and high light intensities, whereas aflibercept moderately enhanced b-wave amplitude only at the high light intensity.

Inner retinal function was further assessed through pattern electroretinogram (PERG) at P21 and P42, which demonstrated no statistical significance among all groups (Appendix A). Intraocular pressure (IOP) was assessed pre- and post-injection with no change detected (Appendix A). In addition, the long-term (P42) effect of anti-Scg3hAb was also evaluated by optical coherence tomography (OCT), which showed an increase in inner retinal thickness in the anti-Scg3hAb group when compared with the aflibercept group but no significance when compared to the PBS group (Appendix A). Outer retinal thickness did not show a significant difference between any of the groups (data did not show). These results are consistent with reduced retinal thickness in aflibercept-treated neonatal mice [4] but opposite to increased foveal thickness in bevacizumab-treated ROP infants [14,15].

### 2.3. Adverse Effects on Visual Function of Healthy Neonatal Mice

To determine the side effects of anti-angiogenic therapy on developing retinas, we intravitreally injected anti-Scg3 hAb and aflibercept into healthy C57BL6/J mice. Short-term (P21) and long-term (P42) adverse effects were assessed using pfERG (Figure 4). Similar trends were observed for both short-term and long-term post-treatment durations, wherein aflibercept decreased a-wave and b-wave amplitudes as well as increased latency of b-waves but not a-waves (not shown), suggesting that the anti-VEGF therapy is detrimental to retinal function. In contrast, anti-Scg3 hAb was without effects on any ERG parameters, including the amplitudes and latencies of a- and b-waves, compared to PBS or no injection control groups.

### 2.4. Adverse Systemic Effects in Neonatal Mice

To evaluate whether systemic anti-angiogenic therapy adversely affects the development of retinal neurons, we injected i.p. anti-Scg3 hAb, aflibercept or PBS into healthy C57BL6/J mice (10 mg/kg) every other day from P7 to P19 and assessed ocular functions by pfERG at P21. Aflibercept treatment had a deleterious effect on the developing retina, which was detected as a significant decrease in a-wave and b-wave amplitudes and significantly increased b-wave latency. However, we observed no changes of any pfERG parameter in parallel anti-Scg3 hAb-treated mice compared to PBS control mice (Figure 5). These findings suggest that systemic aflibercept confers adverse side effects on retinal neuron function.

We further evaluated whether systemic anti-angiogenic therapy confers adverse effects on the global development of neonatal mice (Figure 6). Mice treated with aflibercept via i.p. injection starting at P7 sustained reduced body weight gain at all subsequent time points and weight loss after P15. Conversely, body weights of mice treated with anti-Scg3 hAb were no different from control PBS treatments (Figure 6A,B). Aflibercept treatment conferred significantly decreased kidney size that included loss of density and numbers of kidney endothelial cells, whereas kidneys and resident endothelium of the anti-Scg3 hAb treatment group were normal (Figure 6C–E). Histological analyses revealed abnormal convoluted tubules and glomeruli of kidneys in the aflibercept treatment group, whereas kidneys of mice in the anti-Scg3 hAb group were normal when compared to PBS control (Figure 6F).

## 3. Discussion

ROP is a multifactorial disease associated with vaso-obliteration and subsequent aberrant vaso-proliferation. The disease is currently treated by cryotherapy or laser photocoagulation, both of which can cause retinal damage with loss of the peripheral vision to preserve the central vision. There are presently no approved drug therapies for ROP in the U.S. VEGF inhibitors are often used off-label to treat the disease but have potential safety concerns because of adverse effects on the developing retina and brain in neonates [2,3,7,8,9,10,11,16,17,18].

By applying a novel comparative ligandomics profiling technique, our group recently discovered Scg3 as a unique disease-restricted angiogenic factor [13]. We subsequently validated the potential therapeutic applications of anti-Scg3 mAbs in various animal models of ocular diseases [6,13]. In the current study, we compared the efficacy and safety of a full-length anti-Scg3 hAb with those of aflibercept in a mouse model of ROP. Our results show that anti-Scg3 hAb treatment not only reduced pathological RNV but also facilitated revascularization of the central avascular retina (vaso-obliteration), both of which represent therapeutic benefits for OIR. In contrast, aflibercept ameliorated only OIR-induced RNV while failing to impact revascularization in the avascular area. These dual effects of anti-Scg3 versus aflibercept are consistent with our previous findings for the anti-Scg3 ML78.2 mAb [6], whereas a separate study from our group showed that both anti-Scg3 ML49.3 mAb and aflibercept neither enlarged nor reduced the central avascular area [13].

Anti-VEGF drugs have been extensively investigated in various animal models of ROP and were reported to increase, decrease, or have no effects on the avascular area by different groups [4,5,13,19,20,21]. These conflicting results of VEGF inhibition on vaso-obliteration may result from differences in animal species, strains, OIR models, drug delivery schedule, different anti-VEGF agents, and injection techniques, and they are yet to be reconciled. However, under identical experimental conditions conducted in parallel time courses, the studies reported here clearly show that anti-Scg3 hAb but not aflibercept significantly reduced the central avascular area of OIR mice, indicating therapeutic benefit by facilitating revascularization of the former but not latter treatment.

In addition, disease represents an ominous sign of severe ROP and is characterized by arterial tortuosity and venous dilation [22]. Arterial tortuosity refers to the abnormal twists and curves of arteries in the retina. Although the causes of such vascular abnormalities are incompletely understood, they have variously been associated with changes of blood flow and pressure, malfunctional vessel valves, endothelial dysfunction and hypoxia, and high local VEGF concentration [22,23,24]. To assess such plus disease-like symptoms in OIR mice, we applied in vivo FA at P18 and found that both anti-Scg3 hAb and aflibercept are equally effective to reduce OIR severity by ameliorating arterial tortuosity and venous dilation.

ERG is widely used to assess retinal function. pfERG uses a high-intensity light flash to stimulate photoreceptors (mainly cones) and measure photoreceptor function and downstream retinal signal transmission through the inner plexiform layer. In this study, we chose the light intensity at 20.0 cd·s/m^2^, under which rod photoreceptor activities are largely suppressed, while cone activities are minimally suppressed [25]. sfERG takes advantage of dark adaptation to suppress cone activities and uses a flash with low light intensity to selectively stimulate rod photoreceptors. Dark adaptation is a critical step for achieving maximal rod sensitivity and maintaining cone stimulation at a minimum, and therefore, it is important to adapt animals for a minimum of 16–18 h [26,27]. Insufficient dark adaptation can result in cone signal interference that masks signals contributed by rods.

Symptoms of OIR in the mouse model undergo time-dependent natural regression [28,29], such that differences in retinal function between OIR and healthy age-matched animals remain detectable by ERG in OIR mice at postnatal week 4 but not at week 8 [30]. Based on this disease course, we measured retinal function by ERG at P21 (short-term) and P42 (long-term). The analyses of both pfERG and sfERG confirmed that anti-Scg3 hAb significantly improved overall short- and long-term retinal functions (Figure 3). Differential effects in pfERG and sfERG might result from different rod and cone densities and/or distribution in mice [31]. Rods and cones are heterogeneously distributed in the murine retina with a relatively higher density of cones in the central retina, which is an area affected by OIR.

Aflibercept therapy decreased both short- and long-term pfERG retinal functions (b-wave amplitude in Figure 3B,E), which may be caused by disruptions of photoreceptor synaptic transmission, aberrant mitochondrial morphology, or cellular apoptosis [32,33,34]. No significant adverse effect of aflibercept was observed on short- or long-term a-wave of pfERG and sfERG in OIR mice (Figure 3A,D,G,J). Given that photoreceptor activity is observed through a-wave amplitude, these findings suggest that aflibercept had no therapeutic benefits or adverse effects on photoreceptor function. b-wave analysis captures bipolar signal propagation. A previous study reported that focal ERG parameters (a-wave and b-wave) were severely affected by aflibercept therapy [4]; however, our results using sfERG did not reproduce this effect. The difference may be due to focal ERG parameters, in which only a specific area of the stimulated retina is affected by OIR. Based on our results, we expected that inhibition of aflibercept on b-wave (Figure 3B,E) would reduce the activity of retinal ganglion cells (RGCs). However, the measurement of RGC activity by PERG revealed no significant difference between anti-Scg3 hAb, aflibercept, and PBS control groups (Appendix A). This may be due to the complex divergence and integration of neuronal signals.

We further compared the safety of anti-Scg3 hAb and aflibercept in healthy neonatal mice. The former showed no adverse effects on retinal function, whereas the latter significantly reduced short- and long-term retinal function (Figure 4). These results are consistent with our previous study [6]. To further investigate the safety of anti-Scg3 hAb therapy, healthy neonatal mice were treated i.p. with both test reagents and quantified for changes in body weight and kidney development. No adverse effects on any parameter were detected for anti-Scg3 hAb treatment, but aflibercept retarded body weight gain, induced aberrant kidney morphology, reduced vasculature, and elicited adverse effects on retinal function, which is consistent with previous findings [6,20]. These adverse effects may be consequences of VEGF blockade, because VEGF is essential for vascular development and related organogenesis in neonatal mice. The systemic inhibition of anti-VEGF or its receptor induces vascular regression even in adult mice with mature vasculatures [35].

As a result of the relatively short duration of ROP and long antibody half-life in the vitreous chamber with the blood–retina barrier, the disease typically requires only one intravitreal administration of anti-VEGF antibodies in clinical trials [12,36]. In contrast, systemically delivered antibodies may have a relatively short half-life and need repetitive administrations, as highlighted by systemic anti-VEGF Ab therapy of cancer [37]. Likewise, we intravitreally treated OIR mice with a single dose of anti-Scg3 hAb or aflibercept at P14 but repetitively administrated the reagents i.p. at different time points. However, the precise pharmacokinetics of anti-Scg3 hAb should be characterized before clinical trials.

Overall, our results are consistent with other groups that have demonstrated adverse effects of aflibercept on the developing retina and other organs. In contrast, anti-Scg3 hAb was not only bereft of adverse side effects but also conferred therapeutic benefits in OIR mice by supporting revascularization and improving retinal function, which was consistent with a safer and more effective option for ROP treatment.

There has been intense debate regarding the safety of anti-VEGF therapy for ROP patients. Clinical studies have reported retinal abnormalities, the persistence of peripheral avascular retina, ROP recurrence, retinal detachment, decreased systemic serum VEGF, and underdevelopment of other organ systems in ROP infants who received anti-VEGF therapy [2,3,7,8,9,10,11,16,17,18]. Unfortunately, a lack of consensus on the safety issue has hindered the approval of anti-VEGF for ROP in the U.S., despite the recent approval of ranibizumab in the European Union. The significant safety advantages demonstrated in this study support further development and clinical translation of anti-Scg3 hAb as a new therapy for ROP.

## 4. Materials and Methods

### 4.1. OIR Mouse Model

All animal procedures were approved by the Institutional Animal Care and Use Committee (IACUC) at the University of Miami. The OIR model was generated as previously described [29]. Briefly, C57BL/6J mice at P7 were exposed to 75% oxygen along with nursing mothers in a closed chamber (PRO-OX 110 chamber oxygen controller; Biospherix Ltd., Redfield, NY, USA) for 5 days. Mice were returned to room air at P12 and euthanized at P17 for retinal isolation and analysis.

### 4.2. Anti-Angiogenic Therapy

An optimized anti-Scg3 mAb was converted to hAb by Everglades Biopharma, LLC. Mice were anesthetized by an i.p. injection of a cocktail containing ketamine (75 mg/kg body weight) and xylazine (1.5 mg/kg). Anti-Scg3 hAb, aflibercept (Regeneron Pharmaceuticals, Inc. Tarrytown, NY, USA), or control human IgG (hIgG, Sigma, St. Louis, MO, USA) was injected intravitreally with the same agents for both eyes of OIR mice at P14 (2 µg/0.5 µL/eye). Control groups included PBS or no injection. Pathological RNV was evaluated at P17, and vessel morphology was analyzed on P18. Visual function was evaluated on P21 and P42 by ERG. Therapeutic reagents or controls were blinded-coded before injections or during data quantification and unmasked after analysis.

### 4.3. Histopathology and Immunohistochemistry

Eyes were enucleated from euthanized mice at P17 and fixed in 4% paraformaldehyde for 1 h. Retinas were isolated, stained with AF488-IB4 (10 μg/mL, Thermo Fisher, Waltham, MA, USA) overnight at 4 °C, and analyzed by confocal microscopy. Pathological RNV and vaso-obliteration area were quantified, as described [13,29]. All data are normalized against non-injection control. Kidneys were stained using hematoxylin and eosin (H&E) protocols and imaged using a light microscope.

### 4.4. FA

FA was performed in anesthetized mice at P18, as previously described (Spectralis; Heidelberg Engineering, Heidelberg, Germany) [38]. Briefly, pupils were dilated with a topical drop of 1% tropicamide (Bausch + Lomb, Inc., Tampa, FL, USA), followed by i.p. injection of 10% fluorescein sodium (100 mg/kg) (AK- FLUOR; Akorn, Decatur, IL, USA). FA images were obtained for both eyes at 1–2 min post fluorescein injection. Retinal vein width and retinal arterial tortuosity were quantified manually using MATLAB (Mathworks, Natick, MA, USA) with the method described in a previous study [38].

### 4.5. ERG

PERG and pfERG were recorded utilizing the Jörvec PERG system (Jörvec PERG Visual Stimulation Box, M014760L, Miami, FL, USA). Anesthetized mice were maintained at 37 °C using a heating pad (Physitemp TCAT-2LV controller) with an anal reference thermometer, and eyes were moisturized as needed with a drop of balanced saline solution. PERG parameters were as follows: three consecutive responses of 600 contrast reversals, 10.0 K gain, 1.0 Hz high pass, 100.0 Hz low pass, and 360.0 μV rejection. pfERG parameters were adapted from a previous study and are as follows: three consecutive measurements, strength of 20.0 cd·s/m^2^, and a frequency of 1.0 Hz [25]. PERG amplitude was measured as the difference between the highest peak and the consecutive lowest trough. Latency (i.e., eye response time to light flash) was considered the time at which the highest peak was observed. pfERG a-wave was measured from the baseline signal to the lowest point in the first trough, and b-wave was measured from the trough of a-wave to the following highest peak. The a-wave latency was measured as the time difference from the baseline signal to the lowest point in the first trough, and b-wave latency was measured as the time difference from the baseline signal to the following highest peak.

sfERG was recorded using an UTAS system controlled by EM for Windows software (LKC Technologies, Gaithersburg, MD), as previously described [39]. In brief, mice were dark adapted for 18 h and stimulated 10 times with interstimulus of 5 s, and white light flash was delivered at a range from 0.01 to 1 cd·s/m^2^. Notch filtering was used to reduce the 60 Hz signal noise; low-cut filtering was set at 0.3 Hz; high-cut filtering was set at 500 Hz. The amplitude and latency of a- and b-waves of sfERG were measured, as described above for pfERG.

### 4.6. IOP Measurement

Mice were transiently anesthetized by isoflurane inhalation. IOP measurements were immediately taken using a tonometer (Model TONOLAB TV02, Icare, Vantaa, Finland) with three consecutive measurements recording an average value. Five IOP measurements were taken for all groups at the following timepoints: pre-intravitreal injection, 2 h post-intravitreal injection, 1, 2, and 3 DPI.

### 4.7. OCT

OCT was performed at P42 as previously described (Spectralis™ HRA + OCT device from Heidelberg Engineering, Franklin, MA, USA) [40]. Briefly, set the Automatic Real-Time (ART) value to at least 7 to obtain high-quality images. (The higher the “ART” value, the higher the signal-to-noise ratio and thus the image quality. However, by increasing the “ART” value, the acquiring time is also increased.) B-scans of the mouse retina were obtained with a 55 ° lens. Retinal thickness was analyzed by OCT device’s software.

### 4.8. Systemic Toxicity

Aflibercept, anti-Scg3 hAb, or PBS was injected i.p. (10 mg/kg) into C57BL/6J WT mice (male and female) every other day from P7 to P19. Photopic ERG was performed at P21 in anesthetized mice, followed by euthanasia, kidney isolation, histopathological examinations, and Western blot analyses.

### 4.9. Western Blot

Kidneys were isolated and homogenized on ice using a glass douceur in RIPA buffer (Thermo, 89900). Protein concentration was quantified using the Pierce BCA Protein Assay Kit (Thermo, 23225) following the manufacturer’s recommendations. Equal amounts of protein were resolved using sodium dodecyl sulfate–polyacrylamide gel electrophoresis (SDS-PAGE), transferred to nitrocellulose membranes, and detected by Western blot using rabbit anti-CD31 polyclonal antibodies (Abcam, ab124432) or anti-β-actin mAb (Abcam, ab8226). After washing, bound primary antibodies were detected by horseradish peroxidase-conjugated secondary antibodies and visualized using enhanced chemiluminescence (ECL) Plus Western Blotting Substrate (Thermo, 32134).

### 4.10. Statistical Analysis

Data analysis was performed using the statistical software GraphPad Prism 9 (GraphPad Software, Inc., San Diego, CA, USA). All data are presented as mean ± SEM. One-way analysis of variance (ANOVA) was used to compare differences among groups. Data are considered significant when *p* < 0.05.

## Figures and Tables

**Figure 1 ijms-23-00350-f001:**
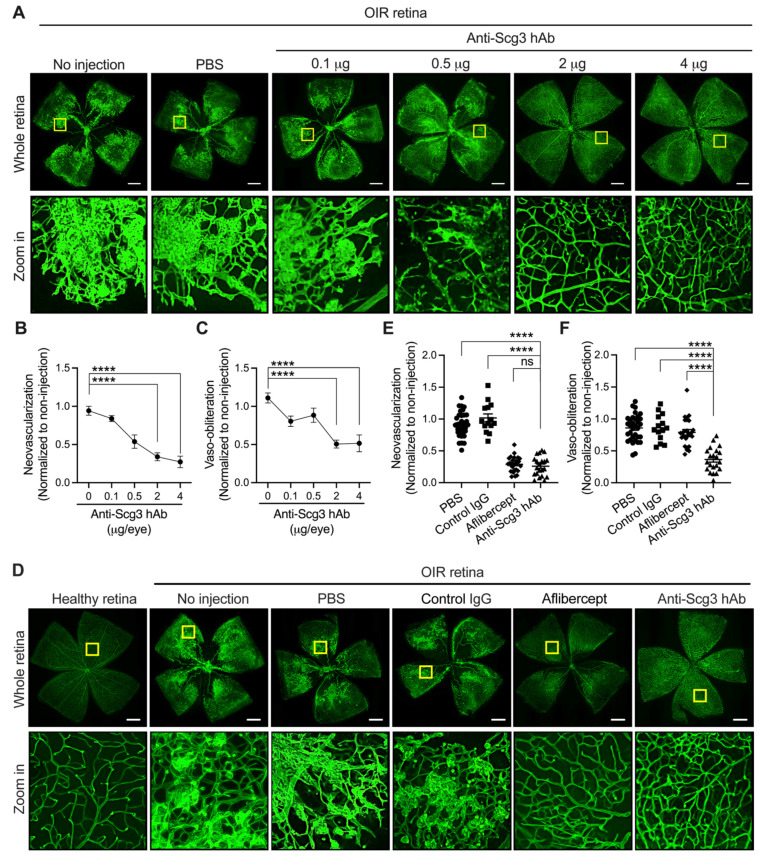
Intravitreal administration of anti-Scg3 hAb inhibits pathological RNV and reduces central avascular area in OIR mice. (**A**) Flat-mount retinas of OIR mice treated with anti-Scg3 hAb at 0.1, 0.5, 2, or 4 µg/0.5 µL/eye. Retinal vessels were stained with AF488-IB4. Yellow box corresponds to the magnified area below, showing changes in vascular pattern. (**B**) Quantification of RNV in (**A**). (**C**) Quantification of the central avascular area in (**A**). *n* = 8 eyes (0, 0.1, and 2 μg) and 6 (0.5 and 4 μg) in (**B**,**C**). (**D**) Flat-mount retina of OIR mice treated with different reagents (2 µg/0.5 µL/eye). (**E**) Quantification of RNV in (**D**). (**F**) Quantification of the central avascular area in (**D**). *n* = 33 eyes (PBS), 14 (control IgG), 26 (aflibercept) and 21 (anti-Scg3 hAb). Mean ± SEM; **** *p* < 0.0001; ns, no significance. one-way ANOVA test. Scale bar = 500 μm (whole retina).

**Figure 2 ijms-23-00350-f002:**
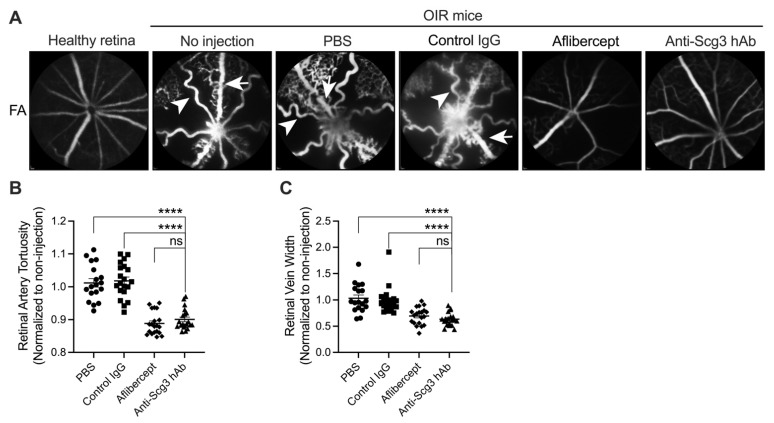
Intravitreal anti-Scg3 hAb ameliorates aberrant retinal artery tortuosity and decreases vein width in OIR mice. (**A**) Representative FA images of OIR mice treated with the indicated agents (2 µg/0.5 µL/eye). Arrowheads indicate tortuous arteries; arrows show dilated veins. (**B**) Quantification of retinal arterial tortuosity in (**A**). Tortuosity index = actual vessel length/linear length. (**C**) Quantification of vein width in (**A**). Mean ± SEM; *n* = 10 eyes (non-injection), 18 (PBS), 20 (control hIgG), 20 (aflibercept), and 23 (anti-Scg3 hAb). **** *p* < 0.0001; ns, not significant; one-way ANOVA test.

**Figure 3 ijms-23-00350-f003:**
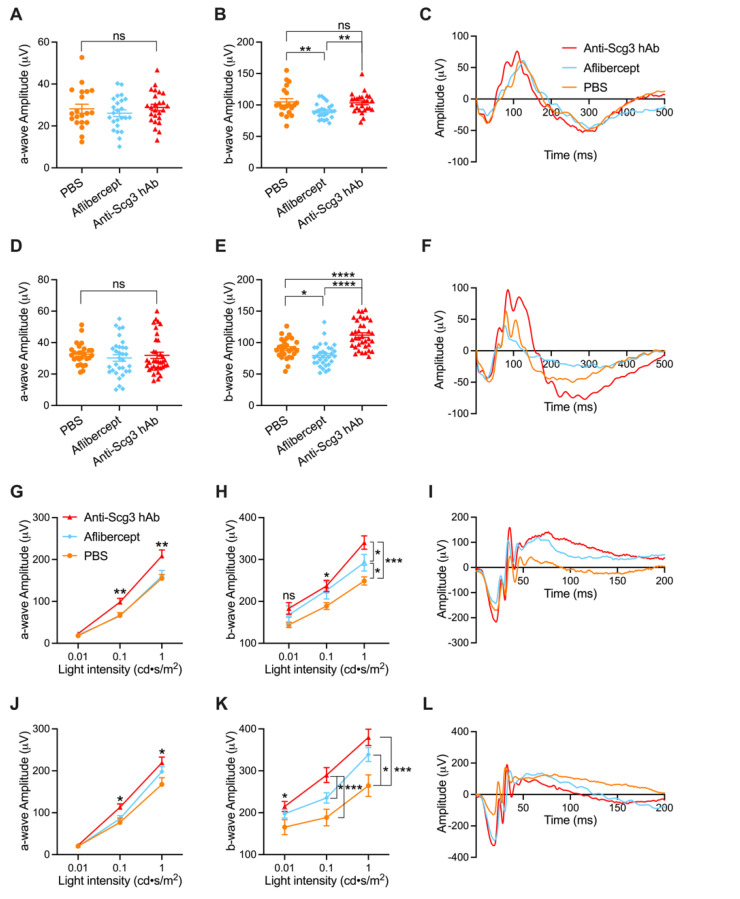
Intravitreal anti-Scg3 hAb improves short- and long-term visual function in OIR mice. (**A**–**C**) Short-term pfERG of OIR mice at P21 (7 DPI). a-wave amplitude remained unchanged between groups (**A**). Intravitreal aflibercept suppressed b-wave amplitude (**B**). *n* = 20 eyes (PBS), 24 (aflibercept), and 25 (anti-Scg3 hAb). (**D**–**F**) Long-term pfERG of OIR mice at P42 (28 DPI). a-wave amplitude was unchanged (**D**). Anti-Scg3 improved b-wave amplitude, whereas aflibercept decreased the amplitude (**E**). *n* = 27 eyes (PBS), 32 (aflibercept), 36 eyes (anti-Scg3 hAb). (**G**–**I**) Short-term sfERG of OIR mice at P21 (7 DPI). Anti-Scg3 hAb significantly improved both a- and b-wave amplitudes at P21 in (**G**,**H**) with more improvement at higher light intensity. Aflibercept improved only b-wave amplitude at the highest light intensity (**H**). *n* = 14 eyes (PBS), 12 (aflibercept), 10 eyes (anti-Scg3 hAb). (**J**–**L**) Long-term sfERG of OIR mice at P42 (28 DPI). The results are similar to P21. *n* = 12 eyes (PBS), 22 (aflibercept), 22 eyes (anti-Scg3 hAb). (**C**,**F**,**I**,**L**) are representative graphs of ERG in each group. (**I**,**L**) show light intensity at 1 cd·s/m^2^. Mean ± SEM. * *p* < 0.05, ** *p* < 0.01, *** *p* < 0.001, **** *p* < 0.0001; ns, not significant; one-way ANOVA test.

**Figure 4 ijms-23-00350-f004:**
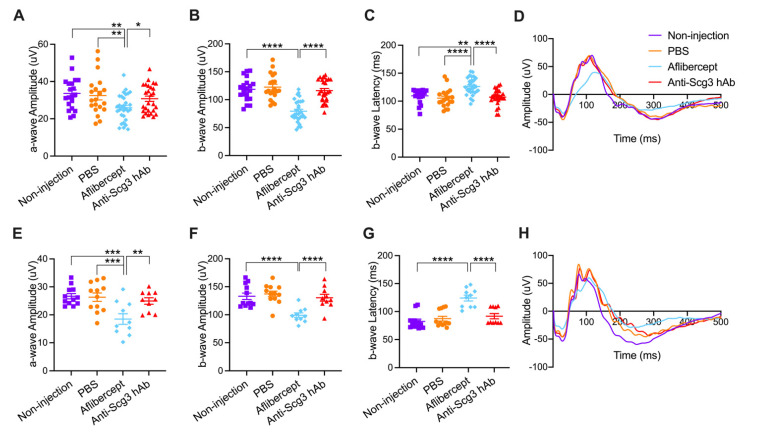
Intravitreal anti-Scg3 hAb does not adversely affect visual function in healthy neonatal mice. (**A**–**D**) Short-term pfERG of healthy mice at P21 (7 DPI). *n* = 20 eyes (non-injection), 20 (PBS), 27 eyes (anti-Scg3 hAb), and 30 (aflibercept). (**E**–**H**) Long-term pfERG of healthy mice at P42 (28 DPI). *n* = 12 eyes (non-injection), 12 (PBS), 10 (anti-Scg3 hAb), and 10 (aflibercept). (**A**,**E**), a-wave amplitude. (**B**,**F**), b-wave amplitude. (**C**,**G**), b-wave latency. (**D**,**H**), average ERG graphs of all mice in each group. Aflibercept decreased a- and b-wave amplitudes at P21 and P42 (**A**,**B**,**E**,**F**). Additionally, aflibercept also increased b-wave latency at P21 and P42 (**C**,**G**). No adverse effects of anti-Scg3 hAb on ERG were detected at either P21 or P42. Mean ± SEM. * *p* < 0.05, ** *p* < 0.01, *** *p* < 0.001, **** *p* < 0.0001; ns, not significant; one-way ANOVA test.

**Figure 5 ijms-23-00350-f005:**
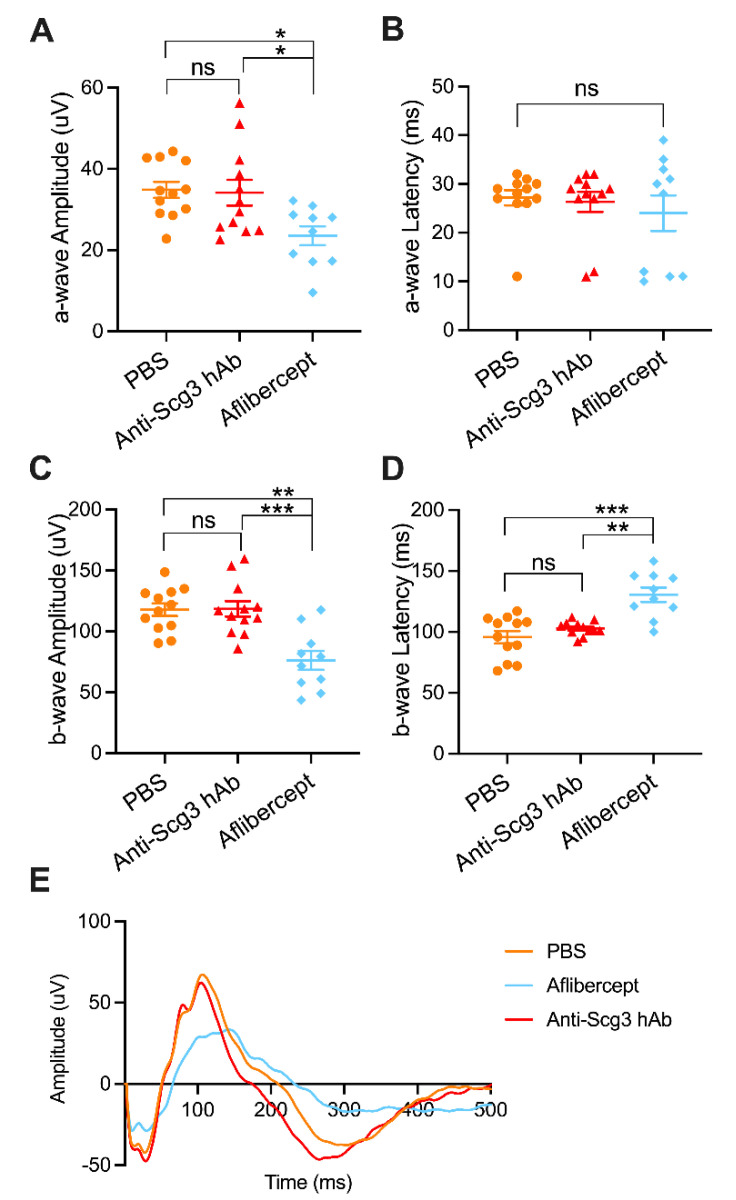
Systemic treatment with anti-Scg3 hAb does not adversely affect visual function in healthy neonatal mice. (**A**–**E**) Short-term pfERG of healthy mice at P21 (i.p. treatment began at P7, followed by continually i.p. injection every other day until P19) detected no change in a-wave amplitude and latency (**A**,**B**) as well as b-wave amplitude and latency (**C**,**D**) with anti-Scg3 hAb. Aflibercept therapy demonstrated a decrease in all parameters except in the latency of a-wave. Average ERG graphs of all mice in each group were depicted in (**E**). Mean ± SEM. *n* = 12 eyes (PBS), 10 eyes (aflibercept) and 12 eyes (anti-Scg3 hAb). * *p* < 0.05, ** *p* < 0.01, *** *p* < 0.001; ns, not significant; one-way ANOVA test.

**Figure 6 ijms-23-00350-f006:**
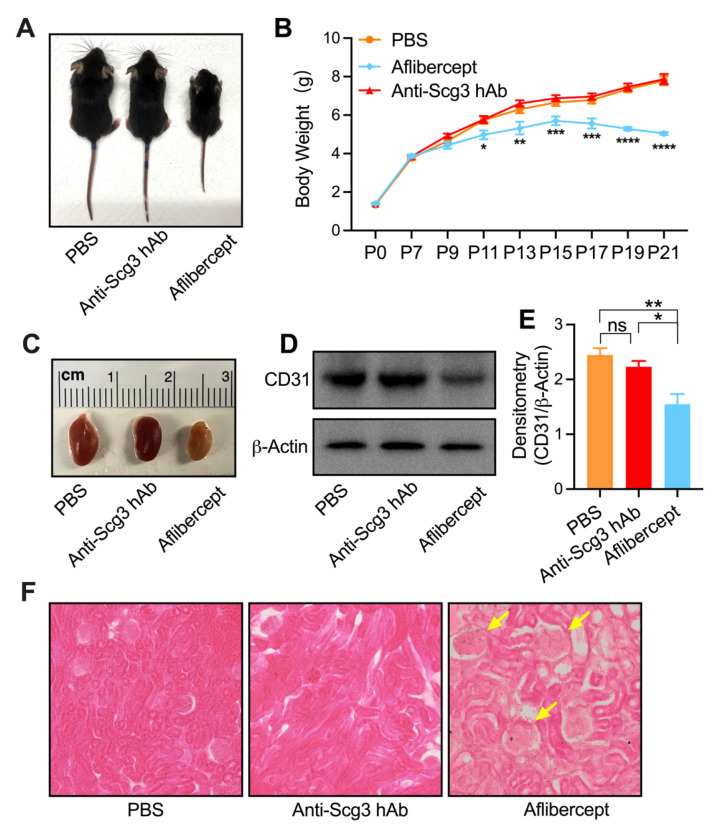
Systemic treatment with anti-Scg3 hAb was without adverse effects on the development of healthy neonatal mice. (**A**) Representative images of body size at P21 for healthy mice treated with PBS, anti-Scg3 hAb or aflibercept (i.p. treatment as in Figure 5A). (**B**) Quantification of the body weight of the mice in A. Aflibercept hinders the body weight gain. *n*= 9 mice (PBS), 10 mice (aflibercept), and 9 mice (anti-Scg3 hAb) for each group. (**C**) Representative images of kidney size at P21 for mice treated with PBS, anti-Scg3 hAb, or aflibercept (i.p. treatment as in Figure 5A). (**D**) Western blot analysis of kidneys in (**C**) using anti-CD31 antibody (endothelial cell marker). Aflibercept reduced kidney endothelial cells. (**E**) Quantification of CD31 signal intensity in (**D**). Data were normalized against β-actin. *n*= 4 kidneys in 4 mice/group. (**F**) Representative image of kidney H&E staining. Arrows indicate aberrant glomerular morphology in the aflibercept group. Mean ± SEM. * *p* < 0.05, ** *p* < 0.01, *** *p* < 0.001, **** *p* < 0.0001; ns, not significant; one-way ANOVA test.

## Data Availability

Data are contained within the article or Appendix A.

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
