# Peer review of "Optimal Efficacy and Safety of Humanized Anti-Scg3 Antibody to Alleviate Oxygen-Induced Retinopathy"

_ijms, 2021, doi:10.3390/ijms23010350_

Round 1

Reviewer 1 Report

The manuscript entitled "Optimal Efficacy and Safety of Humanized Anti-Scg3 Anti-body to Alleviate Oxygen-Induced Retinopathy" uses oxygen-induced retinopathy (OIR) as an animal model for Retinopathy of prematurity (ROP).
I consider that the work, which follows others already published by the same research group, is well written, well structured and that it uses a set of techniques that are adequate to the intended goals. The results are presented in a clear way and do not raise doubts as to their interpretation. 
I only have to point out the fact that it would be interesting if the authors had performed retinal cross sections to evaluate the possible protective effect of Scg3 on the retinal structure alterations known to be induced by OIR.
The authors should also change the legends included in the Y-axis of the ERG graphs, replacing A-wave and B-wave by a-wave and b-wave.

Author Response

We are grateful for the Reviewers’ valuable comments and the opportunity for revision. The following are our point-by-point responses to the comments with specific revisions highlighted in yellow fluorescence in the text.

Reviewer #1

  1. I only have to point out the fact that it would be interesting if the authors had performed retinal cross sections to evaluate the possible protective effect of Scg3 on the retinal structure alterations known to be induced by OIR.

Response: We performed optical coherence tomography (OCT) as Figure S3 and updated the Methods and Results sections accordingly.

  1. The authors should also change the legends included in the Y-axis of the ERG graphs, replacing A-wave and B-wave by a-wave and b-wave.

Response: Revised as suggested.

Reviewer 2

  1. Please correct the following mistypes:

Line 115: “p42”

Line 168: “Aflibercept decreased a- and b-wave amplitudes and P21 and P42 (A, B, E, F).

Response: We corrected both typos at new Line 112 and Line 171.

  1. Please explain what is meant by P21, P42, etc. If this indicates the number of days, why did you choose this abbreviation format?

Response: P21 and P42 refer to postnatal day 21 and 42. We define “postnatal day 14 (P14)" at Line 70.

We hope that above revisions adequately address the Reviewers’ comments.

Sincerely,

Wei Li, Ph.D.

Professor of Ophthalmology

Knights Templar Eye Foundation Presidential Chair in Ophthalmology

Department of Ophthalmology

Cullen Eye Institute, NC205

Baylor College of Medicine

6565 Fannin Street

Houston, TX 77030

USA
Tel: +1-713-798-8885

Reviewer 2 Report

Summary:

In the present paper, the authors present their research results that was carried out in order to demonstrate that a humanized anti-Scg3 antibody is able to alleviate the symptomatology of retinopathy of prematurity and to decelerate the progression of oxygen-induced retinopathy. Furthermore, the authors analysed the adverse effects on visual functions and the systemic side effects of the test substance, and demonstrated the safety of its administration.

The hypothesis is very interesting and current, the study is original and the study design is appropriate. The methods are clearly described and the manuscript is well structured. The authors contribute to this field of research with new and valuable findings, that can be taken into account in clinical trials. Only some minor revisions are recommended before publication.

Observations:

Please correct the following mistypes:

Line 115: “p42”

Line 168: “Aflibercept decreased a- and b-wave amplitudes and P21 and P42 (A, B, E, F).

Please explain what is meant by P21, P42, etc. If this indicates the number of days, why did you choose this abbreviation format?

Author Response

(The authors gave the same response as above.)
